



# An updated version of the global interior ocean biogeochemical data product, GLODAPv2.2021

Siv K. Lauvset [1], Nico Lange [2], Toste Tanhua [2], Henry C. Bittig [3], Are Olsen [4], Alex Kozyr [5], Marta Álvarez [6], Susan Becker [7], Peter J. Brown [8], Brendan R. Carter [9,10], Leticia Cotrim da Cunha [11], Richard A. Feely [10], Steven van Heuven [12], Mario Hoppema [13], Masao Ishii [14], Emil Jeansson [1], Sara Jutterström [15], Steve D. Jones [4], Maren K. Karlsen [4], Claire Lo Monaco [16], Patrick Michaelis [2], Akihiko Murata [17], Fiz F. Pérez [18], Benjamin Pfeil [15], Carsten Schirnick [2], Reiner Steinfeldt [19], Toru Suzuki [20], Bronte Tilbrook [21], Anton Velo [18], Rik Wanninkhof [22], Ryan J. Woosley [23], and Robert M. Key [24]

[1] NORCE Norwegian Research Centre, Bjerknes Centre for Climate Research, Bergen, Norway
[2] GEOMAR Helmholtz Centre for Ocean Research Kiel, Kiel, Germany
[3] Leibniz Institute for Baltic Sea Research Warnemünde, Rostock, Germany
[4] Geophysical Institute, University of Bergen and Bjerknes Centre for Climate Research, Bergen, Norway
[5] NOAA National Centers for Environmental Information, Silver Spring, MD, USA
[6] Instituto Español de Oceanografía, IEO-CSIC, A Coruña, Spain
[7] UC San Diego, Scripps Institution of Oceanography, San Diego CA 92093, USA
[8] National Oceanography Centre, Southampton, UK
[9] Cooperative Institute for Climate Ocean and Ecosystem Studies, University Washington, Seattle, Washington, USA
[10] Pacific Marine Environmental Laboratory, National Oceanic and Atmospheric Administration, Seattle, Washington, USA
[11] Faculdade de Oceanografia/PPG-Oceanografia, Universidade do Estado do Rio de Janeiro, Rio de Janeiro (RJ), Brazil
[12] Centre for Isotope Research, Faculty of Science and Engineering, University of Groningen, Groningen, the Netherlands
[13] Alfred Wegener Institute Helmholtz Centre for Polar and Marine Research, Bremerhaven, Germany
[14] Meteorological Research Institute, Japan Meteorological Agency, Tsukuba, Japan
[15] IVL Swedish Environmental Research Institute, Gothenburg, Sweden
[16] LOCEAN, Sorbonne Université, Paris, France
[17] Research Institute for Global Change, Japan Agency for Marine-Earth Science and Technology, Yokosuka, Japan
[18] Instituto de Investigaciones Marinas, IIM – CSIC, Vigo, Spain
[19] University of Bremen, Institute of Environmental Physics, Bremen, Germany
[20] Marine Information Research Center, Japan Hydrographic Association, Tokyo, Japan
[21] CSIRO Oceans and Atmosphere and Australian Antarctic Program Partnership, University of Tasmania, Hobart, Australia
[22] Atlantic Oceanographic and Meteorological Laboratory, National Oceanic and Atmospheric Administration, Miami, USA.
[23] Center for Global Change Science, Massachusetts Institute for Technology, Cambridge, Massachusetts, USA
[24] Atmospheric and Oceanic Sciences, Princeton University, Princeton, NJ, 08540, USA

*Correspondence to*: siv.lauvset@norceresearch.no



**Abstract.** The Global Ocean Data Analysis Project (GLODAP) is a synthesis effort providing regular compilations of surface-to-bottom ocean biogeochemical bottle data, with an emphasis on seawater inorganic carbon chemistry and related variables determined through chemical analysis of seawater samples. GLODAPv2.2021 is an update of the previous version, GLODAPv2.2020. The major changes are: data from 43 new cruises were added, data coverage extended until 2020, removal of all data with missing temperatures, and the inclusion of a digital object identifier (doi) for each cruise in the product files. In addition, a number of minor corrections to GLODAPv2.2020 data were performed. GLODAPv2.2021 includes measurements from more than 1.3 million water samples from the global oceans collected on 989 cruises. The data for the 12 GLODAP core variables (salinity, oxygen, nitrate, silicate, phosphate, dissolved inorganic carbon, total alkalinity, pH, CFC-11, CFC-12, CFC-113, and $CCl_4$) have undergone extensive quality control with a focus on systematic evaluation of bias. The data are available in two formats: (i) as submitted by the data originator but updated to WOCE exchange format and (ii) as a merged data product with adjustments applied to minimize bias. For this annual update, adjustments for the 43 new cruises were derived by comparing those data with the data from the 946 quality-controlled cruises in the GLODAPv2.2020 data product using crossover analysis. Comparisons to estimates of nutrients and ocean $CO_2$ chemistry based on empirical algorithms provided additional context for adjustment decisions in this version. The adjustments are intended to remove potential biases from errors related to measurement, calibration, and data handling practices without removing known or likely time trends or variations in the variables evaluated. The compiled and adjusted data product is believed to be consistent to better than 0.005 in salinity, 1 % in oxygen, 2 % in nitrate, 2 % in silicate, 2 % in phosphate, 4 $\mu$mol kg$^{-1}$ in dissolved inorganic carbon, 4 $\mu$mol kg$^{-1}$ in total alkalinity, 0.01–0.02 in pH (depending on region), and 5 % in the halogenated transient tracers. The other variables included in the compilation, such as isotopic tracers and discrete $CO_2$ fugacity ($fCO_2$), were not subjected to bias comparison or adjustments.

The original data, their documentation and doi codes are available at the Ocean Carbon Data System of NOAA NCEI (https://www.ncei.noaa.gov/access/ocean-carbon-data-system/oceans/GLODAPv2_2021/, last access: 07 July 2021). This site also provides access to the merged data product, which is provided as a single global file and as four regional ones – the Arctic, Atlantic, Indian, and Pacific oceans – under https://doi.org/10.25921/ttgq-n825 (Lauvset et al., 2021). These bias-adjusted product files also include significant ancillary and approximated data, and can be accessed via www.glodap.info (last access: 29 June 2021). These were obtained by interpolation of, or calculation from, measured data. This living data update documents the GLODAPv2.2021 methods and provides a broad overview of the secondary quality control procedures and results.

## 1 Introduction

The oceans mitigate climate change by absorbing both atmospheric $CO_2$ corresponding to a significant fraction of anthropogenic $CO_2$ emissions (Friedlingstein et al., 2019; Gruber et al., 2019) and most of the excess heat in the Earth System caused by the enhanced greenhouse effect (Cheng et al., 2020; Cheng et al., 2017). The objective of GLODAP (Global Ocean Data Analysis Project, www.glodap.info, last access: 03 June 2021) is to ensure provision of high quality and bias-corrected water column bottle data from the ocean surface to bottom. These data document the state and the evolving changes in physical and chemical ocean properties, e.g., the inventory of the excess $CO_2$ in the ocean, natural oceanic carbon, ocean acidification, ventilation rates, oxygen levels, and vertical nutrient transports (Tanhua et al., 2021). The core quality-controlled and bias-adjusted variables of GLODAP are salinity, dissolved oxygen, inorganic





macronutrients (nitrate, silicate, and phosphate), seawater $CO_2$ chemistry variables (dissolved inorganic carbon – $TCO_2$,

total alkalinity – TAlk, and pH on the total $H^+$ scale), and the halogenated transient tracers chlorofluorocarbon-11 (CFC-11), CFC-12, CFC-113, and $CCl_4$.

Other chemical tracers that are usually measured on the cruises were included in GLODAP. In many cases a subset of these data is distributed as part of the product, however such data have not been extensively quality controlled or checked for measurement biases in this effort. For some of these variables better sources of data exist, for example the product by

85 Jenkins et al. (2019) for helium isotope and tritium data. GLODAP also includes some derived variables to facilitate interpretation, such as potential density anomalies and apparent oxygen utilization (AOU). A full list of variables included in the product is provided in Table 1.

**Table 1.** Variables in the GLODAPv2.2021 comma separated (csv) product files, their units, short and flag names, and corresponding

names in the individual cruise exchange files. In the MATLAB product files that are also supplied a "G2" has been added to every variable name.

| Variable | Units | Product file name | WOCE flag name[a] | 2nd QC flag name[b] | Exchange file name |
|---|---|---|---|---|---|
| Assigned sequential cruise number | | cruise | | | |
| Basin identifier | | region | | | |
| Station | | station | | | STNNBR |
| Cast | | cast | | | CASTNO |
| Year | | year | | | DATE |
| Month | | month | | | DATE |
| Day | | day | | | DATE |
| Hour | | hour | | | TIME |
| Minute | | minute | | | TIME |
| Latitude | | latitude | | | LATITUDE |
| Longitude | | longitude | | | LONGITUDE |
| Bottom depth | m | bottomdepth | | | |
| Pressure of the deepest sample | dbar | maxsampdepth | | | DEPTH |
| Niskin botttle number | | bottle | | | BTLNBR |
| Sampling pressure | dbar | pressure | | | CTDPRS |
| Sampling depth | m | depth | | | |
| Temperature | °C | temperature | | | CTDTMP |
| potential temperature | °C | theta | | | |
| Salinity | | salinity | salinityf | salinityqc | CTDSAL/SALNTY |
| Potential density anomaly | kg m$^{-3}$ | sigma0 | (salinityf) | | |
| Potential density anomaly, ref 1000 dbar | kg m$^{-3}$ | sigma1 | (salinityf) | | |
| Potential density anomaly, ref 2000 dbar | kg m$^{-3}$ | sigma2 | (salinityf) | | |
| Potential density anomaly, ref 3000 dbar | kg m$^{-3}$ | sigma3 | (salinityf) | | |
| Potential density anomaly, ref 4000 dbar | kg m$^{-3}$ | sigma4 | (salinityf) | | |



| Variable | Units | Product file name | WOCE flag name[a] | 2nd QC flag name[b] | Exchange file name |
|---|---|---|---|---|---|
| Neutral density anomaly | kg m$^{-3}$ | gamma | (salinityf) | | |
| Oxygen | µmol kg$^{-1}$ | oxygen | oxygenf | oxygenqc | CTDOXY/OXYGEN |
| Apparent oxygen utilization | µmol kg$^{-1}$ | aou | aouf | | |
| Nitrate | µmol kg$^{-1}$ | nitrate | nitratef | nitrateqc | NITRAT |
| Nitrite | µmol kg$^{-1}$ | nitrite | nitritef | | NITRIT |
| Silicate | µmol kg$^{-1}$ | silicate | silicatef | silicateqc | SILCAT |
| Phosphate | µmol kg$^{-1}$ | phosphate | phosphatef | phosphateqc | PHSPHT |
| TCO$_2$ | µmol kg$^{-1}$ | tco2 | tco2f | tco2qc | TCARBON |
| TAlk | µmol kg$^{-1}$ | talk | talkf | talkqc | ALKALI |
| pH on total scale, 25° C and 0 dbar of pressure | | phts25p0 | phts25p0f | phtsqc | PH_TOT |
| pH on total scale, in situ temperature and pressure | | phtsinsitutp | phtsinsitutpf | phtsqc | |
| $f$CO$_2$ at 20° C and 0 dbar of pressure | µatm | fco2 | fco2f | | FCO2/PCO2 |
| $f$CO$_2$ temperature[c] | °C | $f$co2temp | (fco2f) | | FCO2_TMP/PCO2_TMP |
| CFC-11 | pmol kg$^{-1}$ | cfc11 | cfc11f | cfc11qc | CFC-11 |
| pCFC-11 | ppt | pcfc11 | (cfc11f) | | |
| CFC-12 | pmol kg$^{-1}$ | cfc12 | cfc12f | cfc12qc | CFC-12 |
| pCFC-12 | ppt | pcfc12 | (cfc12f) | | |
| CFC-113 | pmol kg$^{-1}$ | cfc113 | cfc113f | cfc113qc | CFC-113 |
| pCFC-113 | ppt | pcfc113 | (cfc113f) | | |
| CCl$_4$ | pmol kg$^{-1}$ | ccl4 | ccl4f | ccl4qc | CCL4 |
| pCCl$_4$ | ppt | pccl4 | (ccl4f) | | |
| SF$_6$ | fmol kg$^{-1}$ | sf6 | sf6f | | SF6 |
| pSF6 | ppt | psf6 | (sf6f) | | |
| $\delta^{13}$C | ‰ | c13 | c13f | c13qc | DELC13 |
| $\Delta^{14}$C | ‰ | c14 | c14f | | DELC14 |
| $\Delta$14C counting error | ‰ | c14err | | | C14ERR |
| $^3$H | TU | h3 | h3f | | TRITIUM |
| $^3$H counting error | TU | h3err | | | TRITER |
| $\delta^3$He | % | he3 | he3f | | DELHE3 |
| $^3$He counting error | % | he3err | | | DELHER |
| He | nmol kg$^{-1}$ | he | hef | | HELIUM |
| He counting error | nmol kg$^{-1}$ | heerr | | | HELIER |
| Ne | nmol kg$^{-1}$ | neon | neonf | | NEON |
| Ne counting error | nmol kg$^{-1}$ | neonerr | | | NEONER |
| $\delta^{18}$O | ‰ | o18 | o18f | | DELO18 |
| Total organic carbon | µmol L$^{-1\,d}$ | toc | tocf | | TOC |
| Dissolved organic carbon | µmol L$^{-1\,d}$ | doc | docf | | DOC |



| Variable | Units | Product file name | WOCE flag name[a] | 2nd QC flag name[b] | Exchange file name |
|---|---|---|---|---|---|
| Dissolved organic nitrogen | µmol L$^{-1 \, d}$ | don | donf | | DON |
| Dissolved total nitrogen | µmol L$^{-1 \, d}$ | tdn | tdnf | | TDN |
| Chlorophyll $a$ | µg kg$^{-1 \, d}$ | chla | chlaf | | CHLORA |

[a]The only derived variable assigned a separate WOCE flag is AOU as it depends strongly on both temperature and oxygen (and less strongly on salinity). For the other derived variables, the applicable WOCE flag is given in parentheses. [b] Secondary QC flags indicate whether data have been subjected to full secondary QC (1) or not (0), as described in Sect. 3. [c] Included for clarity, is 20 °C for all occurences. [d]Units have not been checked; some values in micromoles per kilogram (for TOC, DOC, DON, TDN) or microgram per liter (for Chl $a$) are probable.

The oceanographic community largely adheres to principles and practices for ensuring open access to research data, such as the FAIR (Findable, Accessible, Interoperable, Reusable) initiative (Wilkinson et al., 2016), but the plethora of file formats and different levels of documentation, combined with the need to retrieve data on a per cruise basis from different access points, limits the realization of their full scientific potential. In addition, the manual data retrieval is time consuming and prone to data handling errors (Tanhua et al., 2021). For biogeochemical data there is the added complexity of different levels of standardization and calibration, and even different units used for the same variable, such that the comparability between data sets is often poor. Standard operating procedures have been developed for some variables (Dickson et al., 2007; Hood et al., 2010; Becker et al., 2019) and certified reference materials (CRM) exist for seawater TCO$_2$ and TAlk measurements (Dickson et al., 2003) and for nutrients in seawater (CRMNS; Aoyama et al., 2012; Ota et al., 2010). Despite this, biases in data still occur. These can arise from poor sampling and preservation practices, calibration procedures, instrument design, and inaccurate calculations. The use of CRMs does not by itself ensure accurate measurements of seawater CO$_2$ chemistry (Bockmon and Dickson, 2015), and the CRMNS have only become available recently and are not universally used. For salinity and oxygen, lack of calibration of the data from conductivity-temperature-depth (CTD) profiler mounted sensors is an additional and widespread problem, particularly for oxygen (Olsen et al., 2016). For halogenated transient tracers, uncertainties in standard gas composition, extracted water volume, and purge efficiency typically provide the largest sources of uncertainty. In addition to bias, occasional outliers occur. In rare cases poor precision - many multiples worse than that expected with current measurement techniques - can render a set of data of limited use. GLODAP deals with these issues by presenting the data in a uniform format, including any metadata either publicly-available or submitted by the data originator, and by subjecting the data to primary and secondary quality control assessments, focusing on precision and consistency, respectively. The secondary quality control focuses on deep data, where natural variability is minimal. Adjustments are applied to the data to minimize cases of bias that could be confidently established relative to the measurement precision for the variables and cruises considered. Key metadata is provided in the header of each data file, and full cruise reports submitted by the data providers are accessible through the GLODAPv2 cruise summary table (https://www.ncei.noaa.gov/access/ocean-carbon-data-system/oceans/GLODAPv2_2021/cruise_table_v2021.html, last access: 07 July 2021).

GLODAPv2.2021 builds on earlier synthesis efforts for biogeochemical data obtained from research cruises, GLODAPv1.1 (Key et al., 2004; Sabine et al., 2005), Carbon dioxide in the Atlantic Ocean (CARINA) (Key et al., 2010), Pacific Ocean Interior Carbon (PACIFICA) (Suzuki et al., 2013), and notably GLODAPv2 (Olsen et al., 2016). GLODAPv1.1 combined data from 115 cruises with biogeochemical measurements from the global ocean. The vast majority of these were the sections covered during the World Ocean Circulation Experiment and the Joint Global Ocean Flux Study (WOCE/JGOFS) in the 1990s, but data from important "historical" cruises were also included, such as from the Geochemical Ocean Sections Study (GEOSECS), Transient Traces in the Ocean (TTO), and South Atlantic





Ventilation Experiment (SAVE). GLODAPv2 was released in 2016 with data from 724 scientific cruises, including those
from GLODAPv1.1, CARINA, PACIFICA, and data from 168 additional cruises. A particularly important source of data
were the cruises executed within the framework of the "repeat hydrography" program (Talley et al., 2016), instigated in
the early 2000s as part of the Climate and Ocean – Variability, Predictability and Change (CLIVAR) program and since
2007 organized as the Global Ocean Ship-based Hydrographic Investigations Program (GO-SHIP) (Sloyan et al., 2019).
GLODAPv2 is now updated regularly using the "living data process" of *Earth System Science Data* to document
significant additions and changes to the dataset.

There are two types of GLODAP updates: full and intermediate. Full updates involve a reanalysis, notably crossover and
inversion, of the entire dataset (both historical and new cruises) and all data points are subject to potential adjustment.
This was carried out for GLODAPv2. For intermediate updates, recently-available data are added following quality
control procedures to ensure their consistency with the cruises included in the latest GLODAP release. Except for obvious
outliers and similar types of errors (Sect. 3.3.1), the data included in previous releases are not changed during
intermediate updates. Additionally, the GLODAP mapped climatologies (Lauvset et al., 2016) are not updated for these
intermediate products. A naming convention has been introduced to distinguish intermediate from full product updates.
For the latter the version number will change, while for the former the year of release is appended. The exact version
number and release year (if appended) of the product used should always be reported in studies, rather than making a
generic reference to GLODAP.

Creating and interpreting inversions, and other checks of the full data set needed for full updates are too demanding in
terms of time and resources to be performed every year or two-years. The aim is to conduct a full analysis (i.e., including
an inversion) again after the third GO-SHIP survey has been completed. This completion is currently scheduled for 2023,
and we anticipate that GLODAPv3 will become available a few years thereafter. In the interim, the third intermediate
update, is presented here which adds data from 43 new cruises to the last update, GLODAPv2.2020 (Olsen et al., 2020).

**2 Key features of the update**

GLODAPv2.2021 contains data from 989 cruises, covering the global ocean from 1972 to 2020, compared to 946 for the
period 1972-2019 for GLODAPv2.2020 (Olsen et al., 2020). Information on the 43 cruises added to this version is
provided in Table A1 in the Appendix. Cruise sampling locations are shown alongside those of GLODAPv2.2020 in Fig.
1, while the coverage in time is shown in Fig. 2. Not all cruises have data for all of the above-mentioned 12 core
variables. For example, cruises with only seawater $CO_2$ chemistry or transient tracer data are still included even without
accompanying nutrient data due to their value towards computation of, for example, carbon inventories. In some other
cases, cruises without any of these properties measured were included – this was because they did contain data for other
carbon related tracers such as carbon isotopes, with the main intention of ensuring their wider availability. The added
cruises are from the years 1982-2020, with most being more recent than 2014. In the Arctic Ocean there are seven cruises
from the Canadian Basin carried out on RV *Louis S. St-Laurent* and one in the Nordic Seas carried out on RV *Johan
Hjort*. In the Pacific Ocean the majority of added cruises are occupations of Line P carried out on RV *John P. Tully*, as
well as a recent occupation of P06 (two legs with different expocodes) on RV *Nathaniel T. Palmer*. Note that for some
Line P cruises only stations with seawater $CO_2$ chemistry data have been included in the product. Thus, all new Pacific
Ocean cruises have seawater $CO_2$ chemistry data. Four out of six cruises added in the Atlantic Ocean (06M220140607
and 06M220160331 on RV *Maria S. Merian* and 06MT20180213 and 06MT20160828 on RV *Meteor*) do not have
seawater $CO_2$ chemistry data, but are included for their transient tracer data. Five new Indian Ocean cruises are added,



including the first occupation of GO-SHIP line I07N since 1995. All new cruises from the Indian Ocean include seawater $CO_2$ chemistry data, including pH on three of them, and transient tracers on all (with the exception of a 1982 cruise in the Red Sea onboard the RV *Marion Dufresne*). Finally, three new cruises are added from the Southern Ocean. All of these include seawater $CO_2$ chemistry.

All new cruises were subjected to primary (Sect. 3.1) and secondary (Sect. 3.2) quality control (QC). These procedures are essentially the same as for GLODAPv2.2020, aiming to ensure the consistency of the data from the 43 new cruises with the previous release of this data product (in this case, the GLODAPv2.2020 adjusted data product).

For GLODAPv2.2021 we have also added a basin identifier to the product files, where 1 is the Atlantic Ocean, 4 is the Arctic Mediterranean Seas, 8 is the Pacific Ocean, and 16 is the Indian Ocean. These regions are abbreviated AO, AMS, PO, and IO respectively in the adjustment table. The basin identifier is now added to the product files to make it easier for users to identify in which ocean basin an individual cruise belongs, without having to use one of the four regional files. In this update we have also included the doi for each cruise in all product files, with the aim of easing access to the original data and metadata as well as improving the visibility of data providers.

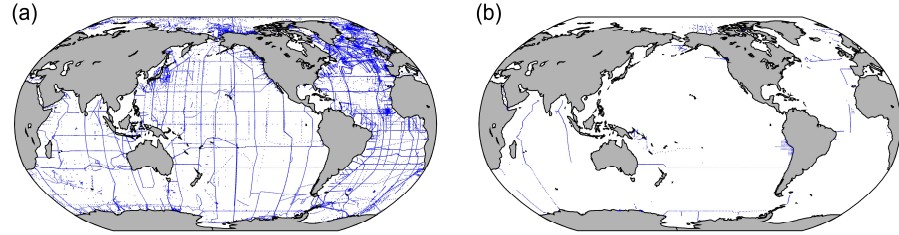

**Figure 1.** Location of stations in (a) GLODAPv2.2020 and for (b) the new data added in this update.

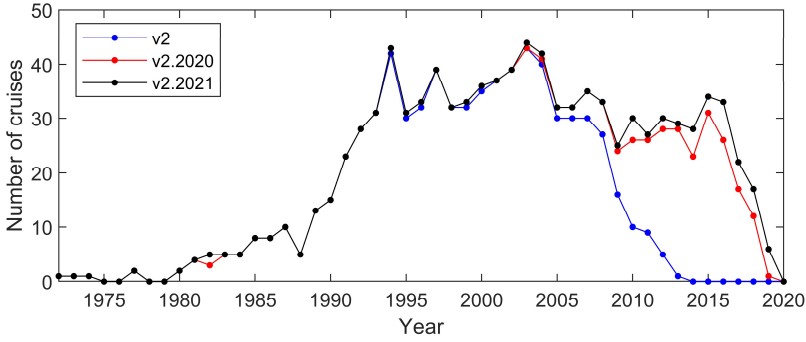

**Figure 2.** Number of cruises per year in GLODAPv2, GLODAPv2.2020, and GLODAPv2.2021.

## 3 Methods

### 3.1 Data assembly and primary quality control

The data from the 43 new cruises were submitted directly to us or retrieved from data centers: typically the CLIVAR and Carbon Hydrographic Data Office (https://cchdo.ucsd.edu, last access: 03 June 2021), National Center for Environmental Information (https://www.ncei.noaa.gov, last access : 03 June 2021), and PANGAEA (https://pangaea.de, last access : 03



June 2021). Each cruise is identified by an expedition code (EXPOCODE). The EXPOCODE is guaranteed to be unique and constructed by combining the country code and platform code with the date of departure in the format YYYYMMDD. The country and platform codes were taken from the ICES (International Council for the Exploration of the Sea) library (https://vocab.ices.dk/, last access 03 June 2021).

The individual cruise data files were converted to the WOCE exchange format: a comma delimited ASCII format for

CTD and bottle data from hydrographic cruises. GLODAP only includes bottle data and CTD data at bottle trip depths, and their exchange format is briefly reviewed here with full details provided in Swift and Diggs (2008). The first line of each exchange file specifies the data type, in the case of GLODAP this is "BOTTLE", followed by a date and time stamp and identification of the group and person who prepared the file, e.g., "PRINUNIVRMK" is Princeton University, Robert M. Key. Next follows the README section; this provides brief cruise specific information, such as dates, ship, region,

method plus quality notes for each variable measured, citation information, and references to any papers that used or presented the data. The README information was typically assembled from the information contained in the metadata submitted by the data originator. In some cases, issues noted during the primary QC and other information such as file update notes are included. The only rule for the README section is that it must be concise and informative. The README is followed by data column headers, units, and then the data. The headers and units are standardized and

provided in Table 1 for the variables included in GLODAP. Exchange file preparation required unit conversion in some cases, most frequently from milliliters per liter (mL $L^{-1}$; oxygen) or micromoles per liter (µmol $L^{-1}$; nutrients) to micromoles per kilogram of seawater (µmol $kg^{-1}$). The default conversion procedure for nutrients was to use seawater density at reported salinity, an assumed measurement-temperature of 22 ºC, and pressure of 1 atm. For oxygen, the factor 44.66 was used for the "milliliters of oxygen" to "micromoles of oxygen" conversion, while the density required for the

"per liter" to "per kilogram" conversion was calculated from the reported salinity and draw temperatures whenever possible. However, potential density was used instead when draw temperature was not reported. The potential errors introduced by any of these procedures are insignificant. Missing numbers are indicated by -999.

Each data column (except temperature and pressure, which are assumed "good" if they exist) has an associated column of data flags. For the original data exchange files, these flags conform to the WOCE definitions for water samples and are

listed in Table 2. For the merged and adjusted product files these flags are simplified: questionable (WOCE flag 3) and bad (WOCE flag 4) data are removed and their flags are set to 9. The same procedure is applied to data flagged 8 (very few such data exist); WOCE flags 1 (Data not received) and 5 (Data not reported) are also set to 9, while flags of 6 (Mean of replicate measurements) and 7 (Manual chromatographic peak measurement) are set to 2, if the data appear good. Also, in the merged product files a flag of 0 is used to indicate a value that could be measured but is approximated: for salinity,

oxygen, phosphate, nitrate, and silicate, the approximation is conducted using vertical interpolation; for seawater $CO_2$ chemistry variables (TCO$_2$, TAlk, pH, and $f$CO$_2$), the approximation is conducted using calculation from two measured $CO_2$ chemistry variables (Sect 3.2.2). Importantly, interpolation of $CO_2$ chemistry variables is never performed and thus a flag value of 0 has a unique interpretation.

If no WOCE flags were submitted with the data, then they were assigned by us. Regardless, all incoming files were

subjected to primary QC to detect questionable or bad data - this was carried out following Sabine et al. (2005) and Tanhua et al. (2010), primarily by inspecting property-property plots. Outliers showing up in two or more different such plots were generally defined as questionable and flagged. In some cases, outliers were detected during the secondary QC; the consequent flag changes have then also been applied in the GLODAP versions of the original cruise data files in agreement with the data submitter.

**Table 2.** WOCE flags in GLODAPv2.2021 exchange format original data files (briefly; for full details see Swift, 2010) and the simplified scheme used in the merged product files.

| WOCE Flag Value | Interpretation | |
|---|---|---|
| | Original data exchange files | Merged product files |
| 0 | Flag not used | Interpolated or calculated value |
| 1 | Data not received | Flag not used[a] |
| 2 | Acceptable | Acceptable |
| 3 | Questionable | Flag not used[b] |
| 4 | Bad | Flag not used[b] |
| 5 | Value not reported | Flag not used[b] |
| 6 | Average of replicate | Flag not used[c] |
| 7 | Manual chromatographic peak measurement | Flag not used[c] |
| 8 | Irregular digital peak measurement | Flag not used[b] |
| 9 | Sample not drawn | No data |

[a]Flag set to 9 in product files

[b]Data are not included in the GLODAPv2.2021 product files and their flags set to 9.

[c]Data are included, but flag set to 2


### 3.2 Secondary quality control

The aim of the secondary QC was to identify and correct any significant biases in the data from the 43 new cruises relative to GLODAPv2.2020, while retaining any signal due to temporal changes. To this end, secondary QC in the form of consistency analyses was conducted to identify offsets in the data. All identified offsets were scrutinized by the

GLODAP reference group through a series of teleconferences during April 2021 in order to decide the adjustments to be applied to correct for the offset (if any). To guide this process, a set of initial minimum adjustment limits was used (Table 3). These are set according to the expected measurement precision for each variable and are the same as those used for GLODAPv2.2020. In addition to the average magnitude of the offsets, factors such as the precision of the offsets, persistence towards the various cruises used in the comparison, regional dynamics, and the occurrence of time trends or

other variations were considered. Thus, not all offsets larger than the initial minimum limits have been adjusted. A guiding principle for these considerations was to not apply an adjustment whenever in doubt. Conversely, in some cases where data and offsets were very precise and the cruise had been conducted in a region where variability is expected to be small, adjustments lower than the minimum limits were applied. Any adjustment was applied uniformly to all values for a variable and cruise, i.e., an underlying assumption is that cruises suffer from either no or a single and constant

measurement bias. Adjustments for salinity, $TCO_2$, TAlk and pH are always additive, while adjustments for oxygen, nutrients and the halogenated transient traces are always multiplicative. Except where explicitly noted (Sect. 3.3.1), adjustments were not changed for data previously included in GLODAPv2.2020.

Crossover comparisons, multi-linear regressions (MLRs), and comparison of deep-water averages were used to identify offsets for salinity, oxygen, nutrients, $TCO_2$, TAlk, and pH (Sect. 3.2.2 and 3.2.3). As in GLODAPv2.2020, but in

contrast to GLODAPv2 and GLODAPv2.2019, evaluation of the internal consistency of the seawater $CO_2$ chemistry variables was not used for the evaluation of pH (Sect. 3.2.4). As in GLODAPv2.2020 we made extensive use of two predictions from two empirical algorithms—"CArbonate system And Nutrients concentration from hYdrological properties and Oxygen using a Neural-network version B" (CANYON-B) and "CONsisTency EstimatioN and amounT" (CONTENT), (Bittig et al., 2018)—for the evaluation of offsets in nutrients and seawater $CO_2$ chemistry data (Section



3.2.5). For the halogenated transient tracers, comparisons of surface saturation levels and the relationships among the tracers were used to assess the data consistency (Sect. 3.2.6). For salinity and oxygen, CTD and bottle values were merged into a "hybrid" variable prior to the consistency analyses (Sect. 3.2.1).

**Table 3.** Initial minimum adjustment limits.

| Variable | Minimum Adjustment |
|---|---|
| Salinity | 0.005 |
| Oxygen | 1 % |
| Nutrients | 2 % |
| $TCO_2$ | 4 µmol kg$^{-1}$ |
| TAlk | 4 µmol kg$^{-1}$ |
| pH | 0.01 |
| CFCs | 5 % |

**3.2.1 Merging of sensor and bottle data**

Salinity and oxygen data can be obtained by analysis of water samples (bottle data) and/or directly from the CTD sensor pack. These two measurement types are merged and presented as a single variable in the product. The merging was conducted prior to the consistency checks, ensuring their internal calibration in the product. The merging procedures were only applied to the bottle data files, which commonly include values recorded by the CTD at the pressures where the

water samples are collected. Whenever both CTD and bottle data were present in a data file, the merging step considered the deviation between the two and calibrated the CTD values if required and possible. Altogether seven scenarios (Table 4) are possible for each of the CTD-O$_2$ sensor properties individually, where the fourth and sixth never occurred during our analyses but is included to maintain consistency with GLODAPv2. The number of cases encountered for each scenario is summarized in Sect. 4.1.

**Table 4.** Summary of salinity and oxygen calibration needs and actions; number of cruises with each of the scenarios identified.

| Case | Description | Salinity | Oxygen |
|---|---|---|---|
| 1 | No data are available: no action needed. | 0 | 1 |
| 2 | No bottle values are available: use CTD values. | 8 | 1 |
| 3 | No CTD values are available: use bottle values. | 2 | 14 |
| 4 | Too few data of both types are available for comparison and >80% of the records have bottle values: use bottle values. | 0 | 0 |
| 5 | The CTD values do not deviate significantly from bottle values: replace missing bottle values with CTD values. | 33 | 23 |
| 6 | The CTD values deviate significantly from bottle values: calibrate CTD values using linear fit and replace missing bottle values with calibrated CTD values. | 0 | 0 |
| 7 | The CTD values deviate significantly from bottle values, and no good linear fit can be obtained for the cruise: use bottle values and discard CTD values. | 0 | 4 |

**3.2.2 Crossover analyses**

The crossover analyses were conducted with the MATLAB toolbox prepared by Lauvset and Tanhua (2015) and with GLODAPv2.2020 as the reference data product. The toolbox implements the 'running-cluster' crossover analysis first



280 described by Tanhua et al. (2010). This analysis compares data from two cruises on a station-by-station basis and calculates a weighted mean offset between the two and its weighted standard deviation. The weighting is based on the scatter in the data such that data that have less scatter have a larger influence on the comparison than data with more scatter. Whether the scatter reflects actual variability or data precision is irrelevant in this context as increased scatter nevertheless decreases the confidence in the comparison. Stations are compared when they are within 2° arc distance (~

285 200 km) of each other. Only deep data are used, to minimize the effects of natural variability. Either the 1500 or 2000 dbar pressure surface was used as upper bound, depending on the number of available data, their variation at different depths, and the region in question. Evaluation was done on a case-by-case basis by comparing crossovers with the two depth limits and using the one that provided the clearest and most robust information. In regions where deep mixing or convection occurs, such as the Nordic, Irminger and Labrador seas, the upper bound was always placed at 2000 dbar;

290 while winter mixing in the first two regions is normally not deeper than this (Brakstad et al., 2019; Fröb et al., 2016), convection beyond this limit has occasionally been observed in the Labrador Sea (Yashayaev and Loder, 2016). However, using an upper depth limit deeper than 2000 dbar will quickly give too few data for robust analysis. In addition, even below the deepest winter mixed layers properties do change over the time periods considered (e.g., Falck and Olsen, 2010), so this limit does not guarantee steady conditions. In the Southern Ocean deep convection beyond 2000 dbar

295 seldom occurs, an exception being the processes accompanying the formation of the Weddell Polynya in the 1970s (Gordon, 1978). Deep and bottom water formation usually occurs along the Antarctic coasts, where relatively thin nascent dense water plumes flow down the continental slope. We avoid such cases, which are easily recognizable. In order to avoid removing persistent temporal trends, all crossover results are also evaluated as a function of time (see below).

 As an example of crossover analysis, the crossover for $TCO_2$ measured on the two cruises 320620170820 (P06E), which

300 is new to this version, and 49NZ20030803, which was included in GLODAPv2, is shown in Fig. 3. For $TCO_2$ the offset is determined as the difference, is in accordance with the procedures followed for GLODAPv2. The $TCO_2$ values from 320620170820 are comparable, with a weighed mean offset of $0.84 \pm 3.12$ µmol kg$^{-1}$ compared to those measured on 49NZ20030803.

 For each of the 43 new cruises, such a crossover comparison was conducted against all possible cruises in

305 GLODAPv2.2020, i.e., all cruises that had stations closer than 2° arc distance to any station for the cruise in question. The summary figure for $TCO_2$ on 320620170820 is shown in Fig. 4. The $TCO_2$ data measured on this cruise are high by $2.15 \pm 1.04$ µmol kg$^{-1}$ when compared to the data measured on nearby cruises included in GLODAPv2.2020. This is well within the initial minimum adjustment limit for $TCO_2$ of 4 µmol kg$^{-1}$ (Table 3), and as such does not qualify for an adjustment of the data in the merged data product. All other variables show the same high consistency (not shown), thus,

310 no adjustment is given to any variable on cruise 320620170820 in GLODAPv2.2021. This is supported by the CANYON-B and CONTENT results (Sect. 3.2.5). Note that adjustments, when applied, are typically round numbers relative to the precision of the variable being considered (e.g., -3 not -3.4 for $TCO_2$ and 0.005 not 0.0047 for pH) to avoid the communicating that the ideal adjustments are known to high precision.



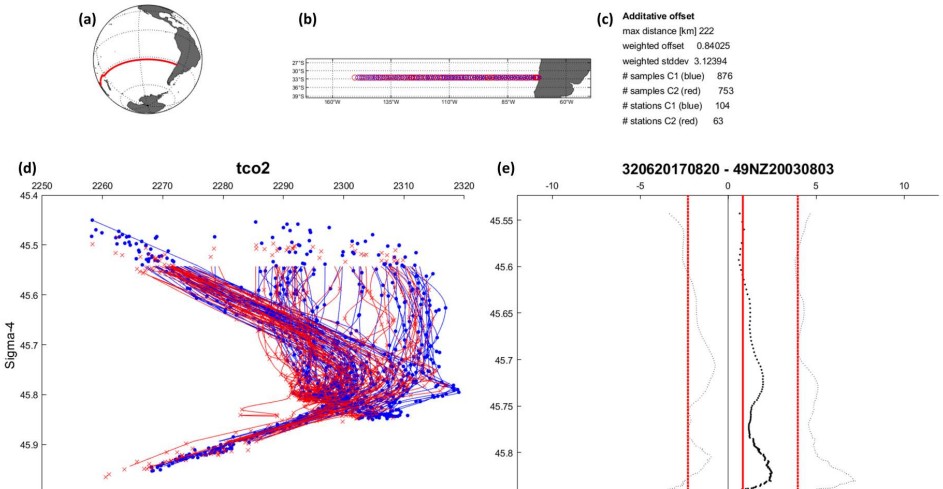

**Figure 3.** Example crossover figure, for TCO₂ for cruises 320620170820 (blue) and 49NZ20030803 (red), as it was generated during the crossover analysis. Panel **(a)** shows all station positions for the two cruises and **(b)** shows the specific stations used for the crossover analysis. Panel **(d)** shows the data of TCO₂ (µmol kg⁻¹) below the upper depth limit (in this case 2000 dbar) versus potential density anomaly referenced to 4000 dbar, as points and the interpolated profiles as lines. Non-interpolated data either did not meet minimum depth separation requirements (Table 4 in Key et al., 2010) or are the deepest sampling depth. The interpolation does not extrapolate. Panel **(e)** shows the mean TCO₂ (µmol kg⁻¹) difference profile (black, dots) with its standard deviation, and also the weighted mean offset (straight, red) and weighted standard deviation. Summary statistics are provided in **(c)**.

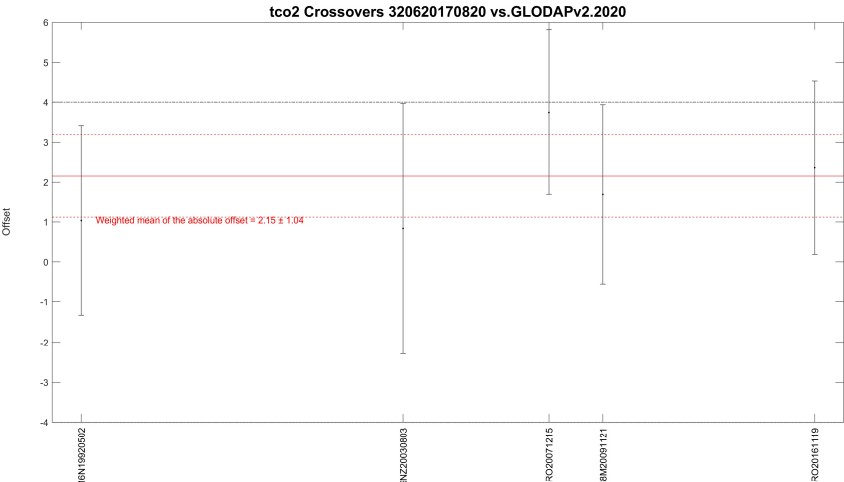

**Figure 4.** Example summary figure, for TCO₂ crossovers for 320620170820 versus the cruises in GLODAPv2.2020 (with cruise EXPOCODE listed on x-axis sorted according to year the cruise was conducted). The black dots and vertical error bars show the weighted mean offset and standard deviation for each crossover (in µmol kg⁻¹). The weighted mean and standard deviation of all these offsets are shown in the red lines and are $2.15 \pm 1.04$ µmol kg⁻¹. The black dashed line is the reference line for a +4 µmol kg⁻¹ offset (the corresponding line for − 4 µmol kg⁻¹ offset is right on top of x-axis and not visible).

### 3.2.3 Other consistency analyses

MLR analyses and deep water averages, broadly following Jutterström et al. (2010), were additionally used for the
secondary QC of salinity, oxygen, nutrients, $TCO_2$, and TAlk data. These approaches are particularly valuable when a
cruise has either very few or no valid crossovers with GLODAPv2; they are used more generally to provide insight on the
consistency of the data. For the 43 new cruises of the present update, no adjustment decisions were made on the basis of
MLR and deep water average analyses alone. The presence of bias in the data was identified by comparing the MLR-
generated values with the measured values. Both analyses were conducted on samples collected deeper than the 1500 or
2000 dbar pressure level to minimize the effects of natural variations, and both used available GLODAPv2.2020 data
from within 2° of the cruise in question to generate the MLR or deep water average. The lower depth limit was set to the
deepest sample for the cruise in question. For the MLRs, all of the above-mentioned variables could be included among
the independent variables (e.g., for a TAlk MLR, salinity, oxygen, nutrients, and $TCO_2$ were allowed), with the exact
selection determined based on the statistical robustness of the fit, as evaluated using the coefficient of determination ($r^2$)
and root mean square error (RMSE). MLRs based on variables that were suspect for the cruise in question were avoided
(e.g., if oxygen appeared biased it was not included as an independent variable). The MLRs could be based on 10 to 500
samples, and the robustness of the fit ($r^2$, RMSE) and quantity of fitting data were considered when using the results to
guide whether to apply a correction. The same applies for the deep-water averages (i.e., the standard deviation of the
mean). MLR and deep-water average results showing offsets above the minimum adjustment limits were carefully
scrutinized, along with available crossover values and CANYON-B and CONTENT estimates, to determine whether or
not to apply an adjustment.

### 3.2.4 pH scale conversion and quality control

Altogether 13 of the 43 new cruises included measured pH data, and none required adjustment (Sect. 4.2). All new pH
data were reported on the total scale and at 25 °C so no scale and/or temperature conversion was necessary. For details on
scale and temperature conversions in previous versions of GLODAPv2 we refer the reader to Olsen et al. (2020). In
contrast to past quality control of GLODAP pH data, evaluation of the internal consistency of $CO_2$ system variables was
not used for the secondary quality control of the pH data of the 13 new cruises; only crossover analysis was used,
supplemented by CONTENT and CANYON-B comparisons (Sect. 3.2.5). Recent literature has demonstrated that internal
consistency evaluation procedures are subject to errors owing to incomplete understanding of the thermodynamic
constants, major ion concentrations, measurement biases, and potential contribution of organic compounds or other
unknown protolytes to alkalinity. These complications lead to pH dependent offsets in calculated pH with cruise
spectrophotometric pH measurements (Álvarez et al., 2020; Carter et al., 2018; Fong and Dickson, 2019), but not with
those derived in lab conditions using ISFET (Ion Sensitive Field Effect Transistor) sensors (Takeshita et al., 2020). The
pH dependent offsets may be interpreted as biases and generate false corrections. The offsets are particularly strong at pH
levels below 7.7, when calculated and measured pH are different by on average between 0.01 and 0.02 units. For the
North Pacific this is a problem as pH values below 7.7 can occur at the depths interrogated during the QC (>1500 dbar for
this region, Olsen et al., 2016). Since any correction, which may be an artifact, would be applied to the full profiles, we
assign an uncertainty of 0.02 to the North Pacific pH data in the merged product files. Elsewhere, the uncertainties that
may have arisen are smaller, since deep pH is typically larger than 7.7 (Lauvset et al., 2020), and at such levels the
difference between calculated and measured pH is less than 0.01 on average (Álvarez et al., 2020; Carter et al., 2018).
Outside the North Pacific, we believe, therefore that the pH data are consistent to 0.01. Avoiding interconsistency



considerations for these intermediate products helps to reduce the problem, but since the reference data set (as also used for the generation of the CANYON-B and CONTENT algorithms) has these issues, a full re-evaluation, envisioned for future GLODAPv3, is needed to address the problem completely.

### 3.2.5 CANYON-B and CONTENT analyses

CANYON-B and CONTENT (Bittig et al., 2018) were used to support decisions regarding application of adjustments (or not). CANYON-B is a neural network for estimating nutrients and seawater $CO_2$ chemistry variables from temperature, salinity, and oxygen concentration. CONTENT additionally considers the consistency among the estimated $CO_2$ chemistry variables to further refine them. These approaches were developed using the data included in the GLODAPv2 data product (i.e., the 2016 version without any more recent updates). Their advantage compared to crossover analyses for evaluating consistency among cruise data is that effects of water mass changes on ocean properties are represented in the non-linear relationships in the underlying neural network. For example, if elevated nutrient values measured on a cruise are not due to a measurement bias, but actual aging of the water masses that have been sampled and as such accompanied by a decrease in oxygen concentrations, the measured values and the CANYON-B estimates are likely to be similar. Vice-versa, if the nutrient values are biased, the measured values and CANYON-B predictions will be dissimilar.

Used in the correct way and with caution this tool is a powerful supplement to the traditional crossover analyses which form the basis of our analyses. Specifically, we gave no weight to comparisons where the crossover analyses had suggested that the S and/or $O_2$ data were biased as this would lead to error in the predicted values. We also considered the uncertainties of the CANYON-B and CONTENT estimates. These uncertainties are determined for each predicted value, and for each comparison the ratio of the difference (between measured and predicted values) to the local uncertainty was used to gauge the comparability. As an example, the CANYON-B/CONTENT analyses of the data obtained for 320620170820 are presented in Fig. 5. The CANYON-B and CONTENT results confirmed the crossover comparisons for $TCO_2$ discussed in Sect. 3.2.2. The magnitude of the inconsistency for both the CONTENT and the CANYON-B estimates was 0.6 µmol kg$^{-1}$, i.e., less than the weighted mean crossover offset of 2.1 µmol kg$^{-1}$ (Fig. 4). The differences between these consistency estimates owes to differences in the actual approach, the weighting across stations, stations considered (i.e., crossover comparisons use only stations within ~200 km of each other, while CANYON-B and CONTENT considers all stations where necessary variables are sampled, and depth range considered (> 500 dbar for CANYON-B and CONTENT vs. >1500/2000 dbar for crossovers). The specific difference between the CANYON-B and CONTENT estimates is a result of the seawater $CO_2$ chemistry considerations by the latter. For the other variables, the inconsistencies are low and agree with the crossover results (not shown here but results can be accessed through the Adjustment Table).

Another advantage of the CANYON-B and CONTENT comparisons is that these procedures provide estimates at the level of individual data points, e.g., pH values are determined for every sampling location and depth where T, S, and $O_2$ data are available. Cases of strong differences between measured and estimated values are always examined. This has helped to identify primary QC issues for some cruises and variables, for example a case of an inverted pH profile on cruise 32PO20130829, which was identified and amended in GLODAPv2.2020.

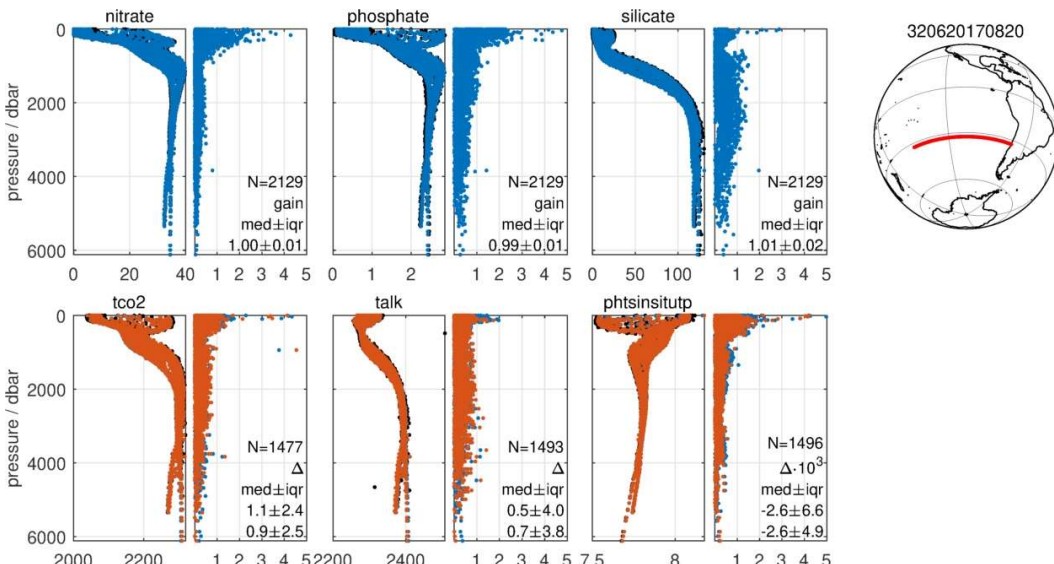

**Figure 5.** Example summary figure for CANYON-B and CONTENT analyses for 320620170820. Any data from regions where CONTENT and CANYON-B were not trained are excluded. The top row shows the nutrients and the bottom row the seawater $CO_2$ chemistry variables. All are shown versus sampling pressure (dbar) and the unit is µmol kg$^{-1}$ for all except pH, which is on the total scale at *in situ* temperature and pressure. Black dots (which to a large extent are hidden by the predicted estimates) are the measured data, blue dots are CANYON-B estimates and red dots are the CONTENT estimates. Each variable has two figure panels. The left shows the depth profile while the right shows the absolute difference between measured and estimated values divided by the CANYON-B/CONTENT uncertainty estimate, which is determined for each estimated value. These values are used to gauge the comparability; a value below 1 indicates a good match as it means that the difference between measured and estimated values is less than the uncertainty of the latter. The statistics in each panel are for all data deeper than 500 dbar and N is the number of samples considered. A multiplicative adjustment and its interquartile range are given for the nutrients. For the seawater $CO_2$ chemistry variables the numbers on each panel are the median difference between measured and predicted values for CANYON-B (upper) and CONTENT (lower). Both are given with their interquartile range.

### 3.2.6 Halogenated transient tracers

For the halogenated transient tracers (CFC-11, CFC-12, CFC-113, and CCl$_4$; CFCs for short) inspection of surface saturation levels and evaluation of relationships between the tracers for each cruise were used to identify biases, rather than crossover analyses. Crossover analysis is of limited value for these variables given their transient nature and low concentrations at depth. As for GLODAPv2, the procedures were the same as those applied for CARINA (Jeansson et al., 2010; Steinfeldt et al., 2010). No QC is performed for SF$_6$ in GLODAP, but there are plans to include this in future versions.

### 3.3 Merged product generation

The merged product file for GLODAPv2.2021 was created by correcting known issues in the GLODAPv2.2020 merged file, and then appending a merged and bias-corrected file containing the 43 new cruises to this error-corrected GLODAPv2.2020 file.

### 3.3.1 Updates and corrections for GLODAPv2.2020

Several minor omissions and errors have been identified in the GLODAPv2.2020 data product since the release in 2020. Most of these have been corrected in this release, but some issues, such as those relating to pH in the North Pacific (Sect.

3.2.4), will not be remedied before GLODAPv3. In addition, some recently available data have been added for a few
cruises. The changes are:

– Individual suspicious samples, identified and reported by users and data providers, have been deleted from the
product. This affects oxygen on cruises 31DS19940126 and 29HE20130320; nutrients on cruises 316N19950829
and 06BE20001128; salinity on cruises 06BE20001128, 316N19921006, 318M19730822, 35A319950221,
49K619940107, and 32PO20130829; and TAlk on cruises 58P320011031, 33RO20071215, and 316N19821201.

– For data with missing (except Gerard bottles, Sect. 3.3.2) or bad temperature all other data have been set to NaN.
For future updates we will attempt to find the missing temperatures and, where possible, restore the now deleted
data

– Corrected all cases where a secondary QC-flag of 1 had been erroneously assigned. This happened for cases where
the secondary QC flag was 1, but the data fields of the entire cruise were only NaN. The only case where this
would be correct is if a -777 is given in the adjustment table; all other cases were changed to a secondary QC-flag
of 0.

– All $f$CO$_2$ data are reported at a constant temperature of 20°C as described in Olsen et al. (2020). In some cases
temperature was not reported for calculated $f$CO$_2$, so where missing, a temperature of 20 °C has been assigned to
calculated $f$CO$_2$ data

– Cruise 18SN19950803 has been given a 8% downward adjustment on phosphate and cruise 49NZ20020822 has
been given a 6% upward adjustment for phosphate. Both were identified as clear outliers when analyzing
crossovers for the seven new cruises in the area (JOIS, Table A1), and the addition of so many new crossovers
allowed for robust assessment of necessary adjustments

– TAlk has been updated for station 106 on cruise 33RO19980123

– Updated data for dissolved total nitrogen (tdn), pH, and TAlk was submitted and included for cruise
33RR20160208. Missing carbon variables have also been calculated for these updated data, and assigned a flag 0

– $\Delta^{14}$C data on 33MW19910711 have been updated

– On cruise 33RO20161119 $\Delta^{14}$C and $\delta^{13}$C data have been added, and BTLNBR updated

– CTDPRS for station 5 (cast 2) on cruise 33RO20131223 have been corrected

**3.3.2 Merging**

The new data were merged into a bias-minimized product file following the procedures used for GLODAPv2.2020 (Olsen
et al., 2020) with some modifications:

– Data from the 43 new cruises were merged and sorted according to EXPOCODE, station, and pressure. GLODAP
cruise numbers were assigned consecutively, starting from 3001, so they can be distinguished from the
GLODAPv2.2020 cruises that ended at 2106.

– For some cruises the combined concentration of nitrate and nitrite was reported instead of nitrate. If explicit nitrite
concentrations were also given, these were subtracted to get the nitrate values. If not, the combined concentration
was renamed to nitrate. As nitrite concentrations are very low in the open ocean, this has no practical implications.

– When bottom depths were not given, they were approximated as the deepest sample pressure +10 dbar or extracted
from ETOPO1 (Amante and Eakins, 2009), whichever was greater. For GLODAPv2, bottom depths were
extracted from the Terrain Base (National Geophysical Data Center/NESDIS/NOAA/U.S. Department of
Commerce, 1995). The intended use of this variable is only drawing approximate bottom topography for sections.



– Whenever temperature was missing in the original data file, all data for that record were removed and their flags set to 9. The same was done when both pressure and depth were missing. For all surface samples collected using buckets or similar, the bottle number was set to zero. There are some exceptions to this, in particular for cruises that also used Gerard barrels for sampling. These may have valuable tracer data that are not accompanied by a temperature, so such data have been retained.

– All data with WOCE quality flags 3, 4, 5, or 8 were excluded from the product files and their flags set to 9. Hence, in the product files a flag 9 can indicate not measured (as is also the case for the original exchange formatted data files) or excluded from the product; in any case, no data value appears. All flags 6 (replicate measurement) and 7 (manual chromatographic peak measurement) were set to 2, provided the data appeared good.

– Missing sampling pressures (depths) were calculated from depths (pressures) following UNESCO (1981).

– For both oxygen and salinity, CTD and bottle values were merged following procedures summarized in Sect. 3.2.1.

– Missing salinity, oxygen, nitrate, silicate, and phosphate values were vertically interpolated whenever practical, using a quasi-Hermetian piecewise polynomial. "Whenever practical" means that interpolation was limited to the vertical data separation distances given in Table 4 in Key et al. (2010). Interpolated salinity, oxygen, and nutrient values have been assigned a WOCE quality flag 0.

– The data for the 12 core variables were corrected for bias using the adjustments determined during the secondary QC.

– Values for potential temperature and potential density anomalies (referenced to 0, 1000, 2000, 3000, and 4000 dbar) were calculated using Fofonoff (1977) and Bryden (1973). Neutral density was calculated using Jackett and McDougall (1997) for all 989 cruises

– Apparent oxygen utilization was determined using the combined fit in Garcia and Gordon (1992).

– Partial pressures for CFC-11, CFC-12, CFC-113, CCl4, and $SF_6$ were calculated using the solubilities by Warner and Weiss (1985), Bu and Warner (1995), Bullister and Wisegarver (1998), and Bullister et al. (2002).

– Missing seawater $CO_2$ chemistry variables were calculated whenever possible. The procedures for these calculations have been slightly altered as the product now contains four such variables; earlier versions of GLODAPv2 (Olsen et al., 2016; Olsen et al., 2019) included only three, so whenever two were included the one to calculate was unequivocal. Four $CO_2$ chemistry variables gives more degrees of freedom in this respect, e.g., a particular record may have measured data for $TCO_2$, TAlk, and pH, and then a choice needs to be made with regard to which pair to use for the calculation of $fCO_2$. We followed two simple principles. First, $TCO_2$ and TAlk was the preferred pair to calculate pH and $fCO_2$, because we have higher confidence in the $TCO_2$ and TAlk data than pH (given the issues summarized in Sect. 3.2.4) and $fCO_2$ (because it was not subjected to secondary QC). Second, if either $TCO_2$ or TAlk was missing and both pH and $fCO_2$ data existed, pH was preferred (because $fCO_2$ has not been subjected to secondary QC). All other combinations involve only two measured variables. The calculations were conducted using CO2SYS (Lewis and Wallace, 1998) for MATLAB (van Heuven et al., 2011), with the carbonate dissociation constants of Lueker et al. (2000), the bisulfate dissociation constant of Dickson (1990), and the borate-to-salinity ratio of Uppström (1974) as in GLODAPv2.2020 and earlier versions (Olsen et al., 2020). We are aware that the borate-to-salinity ratio of Lee et al. (2010) is becoming community standard, but here maintain Uppström (1974) in order to maintain consistency between versions. For calculations involving $TCO_2$, TAlk, and pH, if less than a third of the total number of values, measured and calculated combined, for a



specific cruise were measured, then all these were replaced by calculated values. The reason for this is that secondary QC of the few measured values was often not possible in such cases, for example due to a limited number of deep data available. Such replacements were not done for calculations involving $f\mathrm{CO_2}$, as this would either overwrite all measured $f\mathrm{CO_2}$ values or would entail replacing a measured variable that has been subjected to secondary QC (i.e., $\mathrm{TCO_2}$, TAlk, or pH) with one calculated from a variable that has not been subjected to secondary QC (i.e., $f\mathrm{CO_2}$). Calculated seawater $\mathrm{CO_2}$ chemistry values have been assigned WOCE flag 0. Seawater $\mathrm{CO_2}$ chemistry values have not been interpolated, so the interpretation of the 0 flag is unique.

– The resulting merged file for the 43 new cruises was appended to the merged product file for GLODAPv2.2020.

## 4 Secondary quality control results and adjustments

All material produced during the secondary QC is available via the online GLODAP Adjustment Table hosted by GEOMAR, Kiel, Germany at https://glodapv2-2021.geomar.de/ (last access: 29 June 2021), and which can also be accessed through www.glodap.info. This is similar in form and function to the GLODAPv2 Adjustment Table (Olsen et al., 2016) and includes a brief written justification for any adjustments applied.

### 4.1 Sensor and bottle data merge for salinity and oxygen

Table 4 summarizes the actions taken for the merging of the CTD and bottle data for salinity and oxygen. For 75 % of the 43 new cruises both CTD and bottle data of salinity were included in the original cruise data files and for all these cruises the two data types were found to be consistent. This is similar to the GLODAPv2.2020 results. For oxygen, 63 % of the new cruises included both CTD $\mathrm{O_2}$ and bottle values, which is much more than for GLODAPv2.2020 (25%), but comparable to GLODAPv2.2019. Having both CTD and bottle values in the data files is highly preferred as the information is valuable for quality control (bottle mistrips, leaking Niskin bottles, and oxygen sensor drift are among the issues that can be revealed). The extent to which the bottle data (i.e., OXYGEN in the individual cruise exchange files) is in reality mislabeled CTD data (i.e., should be CTDOXY) is uncertain. Regardless, the large majority of the CTD and bottle oxygen were consistent and did not need any further calibration of the CTD values (23 out of 27 cruises), while for four cruises no good fit could be obtained and their CTD $\mathrm{O_2}$ data are not included in the product.

### 4.2 Adjustment summary

The secondary QC has 5 possible outcomes which are summarized in Table 5, along with the corresponding codes that appear in the online Adjustment Table and that are also occasionally used as shorthand for decisions in the text below. Some cruises could not get full secondary QC. Specifically, in some cases data were too shallow or geographically too isolated for full and conclusive consistency analyses. A secondary QC flag has been included in the merged product files to enable their identification, with "0" used for variables and cruises not subjected to full secondary QC (corresponding to code -888 in Table 5) and "1" for variables and cruises that were subjected to full secondary QC. The secondary QC flags are assigned per cruise and variable, not for individual data points and are independent of—and included in addition to—the primary (WOCE) QC flag. For example, interpolated (salinity, oxygen, nutrients) or calculated ($\mathrm{TCO_2}$, TAlk, pH) values, which have a primary QC flag 0, may have a secondary QC flag of 1 if the measured data these values are based on have been subjected to full secondary QC. Conversely, individual data points may have a secondary QC flag of 0, even if their primary QC flag is 2 (good data). A 0 flag means that data were too shallow or geographically too isolated for


consistency analyses or that these analyses were inconclusive, but that we have no reasons to believe that the data in
question are of poor quality. Prominent examples for this version are the two new cruises in the Salish Sea: no data were
available in this region in GLODAPv2.2020, which, combined with quite shallow sampling depths, rendered conclusive
secondary QC impossible. As a consequence, most, but not all, of these data (some being excluded because of poor
precision after consultation with the PI) are included with a secondary QC flag of 0.

The secondary QC actions for the 12 core variables and the distribution of applied adjustments are summarized in Table 6
and Fig. 6, respectively. For most variables only a small fraction of the data are adjusted: no salinity or pH data, 4.5 % of
$TCO_2$ and TAlk data, 7 % of oxygen data, 14 % of nitrate and phosphate data, and 21 % of silicate data. For the CFCs, no
data required adjustment. Overall, the magnitudes of the various adjustments applied are also small. There is a larger
fraction of data requiring adjustments to nutrients in GLODAPv2.2021 compared to GLODAPv2.2020. However, the
tendency observed during the production of GLODAPv2.2019 and GLODAPv2.2020 remains, namely that the large
majority of recent cruises are consistent with earlier releases of the GLODAP data product.

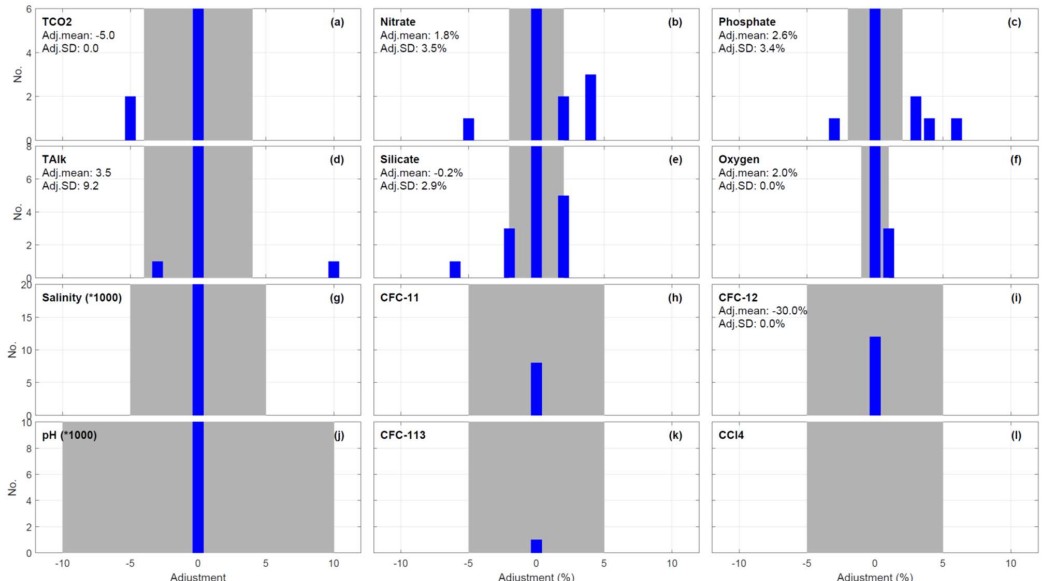

**Figure 6.** Distribution of applied adjustments for each core variable that received secondary QC, in µmol kg⁻¹ for $TCO_2$ and TAlk,
unitless for salinity and pH (but multiplied with 1000 in both cases so a common x-axis can be used), while for the other properties
adjustments are given in percent ((adjustment ratio-1)x100)). Grey areas depict the initial minimum adjustment limits. The figure
includes numbers for data subjected to secondary quality control only. Note also that the y-axis scale is set to render the number of
adjustments to be visible, so the bar showing zero offset (the 0 bar) for each variable is cut off (see Table 6 for these numbers).

**Table 5.** Possible outcomes of the secondary QC and their codes in the online Adjustment Table

| Secondary QC result | Code |
|---|---|
| The data are of good quality, consistent with the rest of the dataset and should not be adjusted. | 0/1[a] |
| The data are of good quality but are biased: adjust by adding (for salinity, $TCO_2$, TAlk, pH) or by multiplying (for oxygen, nutrients, CFCs) the adjustment value | Adjustment value |
| The data have not been QC'd, are of uncertain quality, and suspended until full secondary QC has been carried out | -666 |
| The data are of poor quality and excluded from the data product. | -777 |





| The data appear of good quality but their nature, being from shallow depths, coastal regions, without crossovers or similar, prohibits full secondary QC | -888 |
|---|---|
| No data exist for this variable for the cruise in question | -999 |

[a]The value of 0 is used for variables with additive adjustments (salinity, TCO$_2$, TAlk, pH) and 1 for variables with multiplicative adjustments (for oxygen, nutrients, CFCs). This is mathematically equivalent to 'no adjustment' in both cases

**Table 6.** Summary of secondary QC results for the 43 new cruises, in number of cruises per result and per variable.

| | Sal. | Oxy. | NO$_3$ | Si | PO$_4$ | TCO$_2$ | TAlk | pH | CFC-11 | CFC-12 | CFC-113 | CCl$_4$ |
|---|---|---|---|---|---|---|---|---|---|---|---|---|
| With data | 43 | 42 | 41 | 41 | 40 | 36 | 35 | 13 | 8 | 13 | 1 | 0 |
| No data | 0 | 1 | 2 | 2 | 3 | 7 | 8 | 30 | 35 | 30 | 42 | 43 |
| Unadjusted[a] | 36 | 32 | 27 | 23 | 27 | 28 | 28 | 13 | 8 | 13 | 1 | 0 |
| Adjusted[b] | 0 | 3 | 6 | 9 | 6 | 2 | 2 | 0 | 0 | 0 | 0 | 0 |
| -888[c] | 7 | 7 | 7 | 8 | 7 | 6 | 4 | 0 | 0 | 0 | 0 | 0 |
| -666[d] | 0 | 0 | 0 | 0 | 0 | 0 | 0 | 0 | 0 | 0 | 0 | 0 |
| -777[e] | 0 | 0 | 1 | 1 | 0 | 0 | 0 | 0 | 0 | 0 | 0 | 0 |

[a]The data are included in the data product file as is, with a secondary QC flag of 1.

[b]The adjusted data are included in the data product file with a secondary QC flag of 1.

[c]Data appear of good quality but have not been subjected to full secondary QC. They are included in data product with a secondary QC flag of 0.

[d]Data are of uncertain quality and suspended until full secondary QC has been carried out; they are excluded from the data product.

[e]Data are of poor quality and excluded from the data product.

Only 13 out of the 43 new cruises included measured pH data and none received an adjustment. However, we have not performed a new crossover and inversion analysis of all pH data in the northwestern Pacific (though such analysis is planned for the next full update of GLODAP, i.e., GLODAPv3). Therefore, for now the conclusion from GLODAPv2.2020 remains and some caution should be exercised if looking at trends in ocean pH in the northwestern Pacific using GLODAPv2.2020 or GLODAPv2.2021.

For the nutrients, adjustments were applied to maintain consistency with data included in GLODAPv2, GLODAPv2.2019, and GLODAPv2.2020. An alternative goal for the adjustments would be maintaining consistency with data from cruises that employed CRMNS to ensure accuracy of nutrient analyses. Such a strategy was adopted by Aoyama (2020) for preparation of the Global Nutrients Dataset 2013 (GND13), and is being considered for GLODAP as well. However, as this would require a re-evaluation of the entire data set, this will not occur until the next full update of

GLODAP, i.e., GLODAPv3. For now, we note the overall agreement between the adjustments applied in these two efforts (Aoyama, 2020), and that most disagreements appear to be related to cases where no adjustments were applied in GLODAP. This can be related to the strategy followed for nutrients for GLODAPv2, where data from GO-SHIP lines were considered more accurate than other data (Olsen et al., 2016). CRMNS are used for nutrients on most GO-SHIP lines.

The improvement in data consistency due to the secondary QC process is evaluated by comparing the weighted mean of the absolute offsets for all crossovers before and after the adjustments have been applied. This "consistency improvement" for core variables is presented in Table 7. The data for CFCs were omitted from these analyses for



previously discussed reasons (Sect. 3.2.6). Globally, the improvement is modest. Considering the initial data quality, this result was expected. However, this does not imply that the data initially were consistent everywhere. Rather, for some

regions and variables there are substantial improvements when the adjustments are applied. Silicate in the Atlantic Ocean, for example, shows a considerable improvement and nutrients in general show improvements in almost all regions, including globally.

**Table 7.** Improvements resulting from quality control of the 43 new cruises, per basin and for the global data set. The
numbers in the table are the weighted mean of the absolute offset of unadjusted and adjusted data versus GLODAPv2.2020. *n* is the total number of valid crossovers in the global ocean for the variable in question.

| | ARCTIC | | | ATLANTIC | | | INDIAN | | | PACIFIC | | | GLOBAL | | | *n* |
|---|---|---|---|---|---|---|---|---|---|---|---|---|---|---|---|---|
| | Unadj | | Adj | Unadj | | Adj | Unadj | | Adj | Unadj | | Adj | Unadj | | Adj | (global) |
| **Sal ( x1000)** | 3.0 | => | 3.0 | 4.2 | => | 4.2 | 2.4 | => | 2.4 | 2.5 | => | 2.5 | 2.9 | => | 2.9 | 917 |
| **Oxy (%)** | 0.9 | => | 0.9 | 0.9 | => | 0.8 | 0.8 | => | 0.8 | 1.3 | => | 1.2 | 1.0 | => | 1.0 | 842 |
| **NO₃ (%)** | 1.5 | => | 1.3 | 3.3 | => | 1.4 | 1.0 | => | 1.0 | 1.4 | => | 1.0 | 1.5 | => | 1.1 | 670 |
| **Si (%)** | 4.0 | => | 3.6 | 9.2 | => | 1.8 | 1.5 | => | 1.2 | 1.1 | => | 0.8 | 1.7 | => | 1.2 | 665 |
| **PO₄ (%)** | 3.4 | => | 2.8 | 2.6 | => | 1.7 | 0.7 | => | 0.7 | 2.0 | => | 1.8 | 2.2 | => | 1.8 | 643 |
| **TCO₂ (µmol/kg)** | 3.2 | => | 3.2 | 1.9 | => | 1.9 | 1.9 | => | 1.9 | 2.6 | => | 2.3 | 2.6 | => | 2.4 | 328 |
| **TAlk (µmol/kg)** | 3.0 | => | 3.0 | 5.5 | => | 5.5 | 2.2 | => | 2.2 | 2.9 | => | 2.4 | 3.2 | => | 3.0 | 262 |
| **pH ( x1000)** | NA | => | NA | 4.9 | => | 4.9 | 14.8 | => | 14.8 | 11.0 | => | 11.0 | 9.0 | => | 9.0 | 99 |

The various iterations of GLODAP provide insight into initial data quality covering more than 4 decades. Figure 7
summarizes the applied absolute adjustment magnitude per decade. These distributions are broadly unchanged compared to GLODAPv2.2020 (Fig. 8 in Olsen et al., 2020). Most TCO₂ and TAlk data from the 1970s needed an adjustment, but this fraction steadily declines until only a small percentage is adjusted in recent years. This is encouraging and demonstrates the value of standardizing sampling and measurement practices (Dickson et al., 2007), the widespread use of CRMs (Dickson et al., 2003), application of best practices and standardized procedures, and instrument automation.
The pH adjustment frequency also has a downward trend; however, there remain issues with the pH adjustments and this is a topic for future development in GLODAP, with the support from the OCB Ocean Carbonate System Intercomparison Forum (OCSIF, https://www.us-ocb.org/ocean-carbonate-system-intercomparison-forum/, last accessed: 03 June 2021) working group (Álvarez et al., 2020). For the nutrients and oxygen, only the phosphate adjustment frequency decreases from decade to decade. However, we do note that the more recent data from the 2010s receive the fewest adjustments.
This may reflect recent increased attention that seawater nutrient measurements have received through an operation manual (Becker et al., 2019; Hydes et al., 2012) availability of CRMNS (Aoyama et al., 2012; Ota et al., 2010), and the SCOR working group #147, Towards comparability of global oceanic nutrient data (COMPONUT). For silicate, the fraction of cruises receiving adjustments peaks in the 1990s and 2000s. This is related to the 2 % offset between US and



Earth System
Science
Data

Japanese cruises in the Pacific Ocean that was revealed during production of GLODAPv2 and discussed in Olsen et al.

(2016). For salinity and the halogenated transient tracers, the number of adjusted cruises is small in every decade.

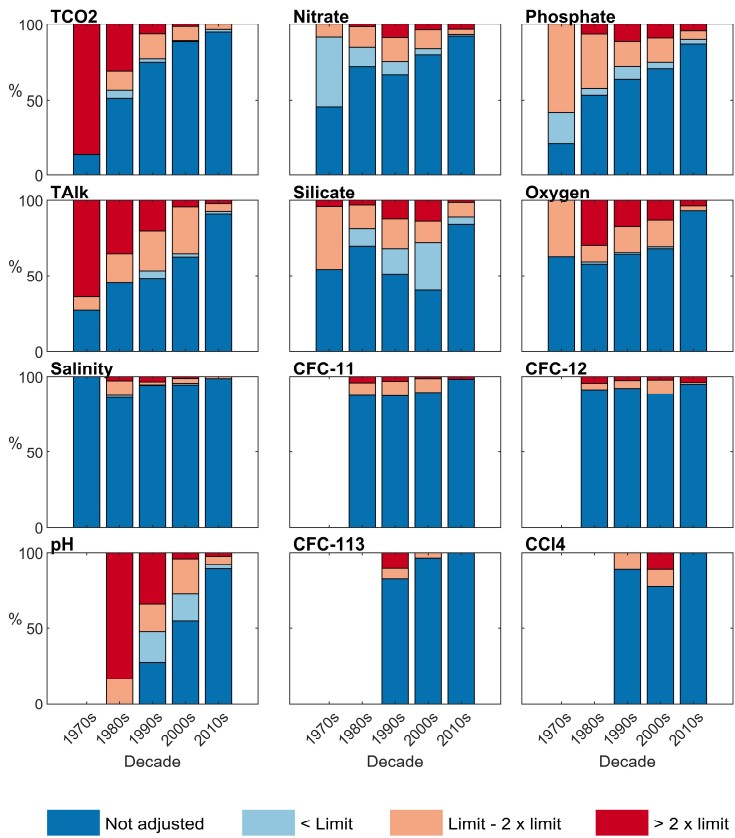

**Figure 7.** Magnitude of applied adjustments relative to minimum adjustment limits (Table 3) per decade for the 989 cruises included in GLODAPv2.2021.

**5 Data availability**

The GLODAPv2.2021 merged and adjusted data product is archived at NOAA NCEI under https://doi.org/10.25921/ttgq-n825 (Lauvset et al., 2021). These data and ancillary information are also available via our web pages https://www.glodap.info and https://www.ncei.noaa.gov/access/ocean-carbon-data-system/oceans/GLODAPv2_2021/ (last access: 07 July 2021). The data are available as comma-separated ascii files (*.csv) and as binary MATLAB files (*.mat) that use the open-source Hierarchical Data Format version 5 (HDF5). The data product is also made available as

an Ocean Data View (ODV) file which can be easily explored using the "webODV Explore" online data service (https://explore.webodv.awi.de/, last access: 07 July 2021). Regional subsets are available for the Arctic, Atlantic, Pacific, and Indian oceans. There are no data overlaps between regional subsets and each cruise exists in only one basin file even if data from that cruise crosses basin boundaries. The station locations in each basin file are shown in Fig. 8. The product file variables are listed in Table 1. A lookup table for matching the EXPOCODE of a cruise with GLODAP cruise number is provided with the data files, and a similar table is provided for matching the GLODAP cruise number with the




data DOI. In the MATLAB files this information (EXPOCODE and DOI) is available as a cell array. A "known issues document" accompanies the data files and provides an overview of known errors and omissions in the data product files. It is regularly updated, and users are encouraged to inform us whenever any new issues are identified. It is critical that users consult this document whenever the data products are used.

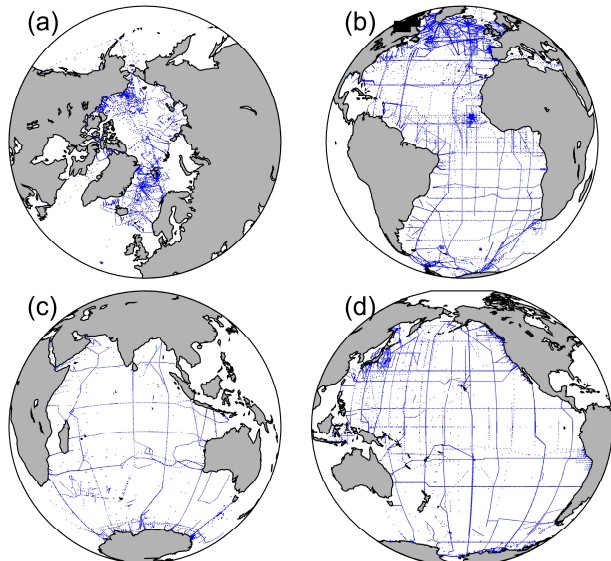


**Figure 8.** Locations of stations included in the (a) Arctic, (b) Atlantic, (c) Indian, and (d) Pacific Ocean product files for the complete GLODAPv2.2021 dataset.

The original cruise files, with updated flags determined during additional primary GLODAP QC, are available through
the GLODAPv2.2021 cruise summary table (CST) hosted by NOAA NCEI: https://www.ncei.noaa.gov/access/ocean-carbon-data-system/oceans/GLODAPv2_2021/cruise_table_v2021.html (last access: 07 July 2021). Each of these files has been assigned a doi, which is included in the data product files, but not listed here. The CST also provides brief information on each cruise and access to metadata, cruise reports, and its Adjustment Table entry.

While GLODAPv2.2021 is made available without any restrictions, users of the data should adhere to the fair data use
principles:

For investigations that rely on a particular (set of) cruise(s), recognize the contribution of GLODAP data contributors by at least citing the articles where the data are described and, preferably, contacting principal investigators for exploring opportunities for collaboration and co-authorship. To this end, relevant articles and principal investigator names are provided in the cruise summary table. Contacting principal investigators comes with the additional benefit that the
principal investigators often possess expert insight into the data and/or specific region under investigation. This can improve scientific quality and promote data sharing.

This paper should be cited in any scientific publications that result from usage of the product. Citations provide the most efficient means to track use, which is important for attracting funding to enable the preparation of future updates.



## 6 Summary

GLODAPv2.2021 is an update of GLODAPv2.2020. Data from 43 new cruises have been added to supplement the earlier release and extend temporal coverage by 1 year. GLODAP now includes 47 years, 1972–2020, of global interior ocean biogeochemical data from 989 cruises.

The total number of data records is 1 334 269. Records with measurements for all 12 core variables (salinity, oxygen, nitrate, silicate, phosphate, $TCO_2$, TAlk, pH, CFC-11, CFC-12, CFC-113, and $CCl_4$) are very rare; only 2029 records

have measured data for all 12 in the merged product file (interpolated and calculated data excluded). Requiring only two out of the four measured seawater $CO_2$ chemistry variables, in addition to all the other core variables, brings the number of available records up to 9231, so this is also very rare. A major limiting factor to having all core variables is the simultaneous availability of data for all four transient tracer species: only 26 137 records have measurements of CFC-11, CFC-12, CFC-113, and $CCl_4$ while 422 029 have data for at least one of these (not considering availability of other core

variables). A total of 423 544 records have measured data for two out of the three $CO_2$ chemistry core variables. The number of measured $f$CO$_2$ data is 33 844; note again that these data were not subjected to quality control. The number of records with measured data for salinity, oxygen, and nutrients is 832 566, while the number of records with salinity and oxygen data is 1 127 477. All of the above numbers concern measured data, not interpolated or calculated values. 2% (27 538) of the total data records do not have salinity. There are several reasons for this, the main one being the inability

to vertically interpolate due to too large separation (Section 3.3.2) between measured samples. Other reasons for missing salinity include salinity not being reported and missing depth or pressure. Note that there are slightly fewer records with $f$CO$_2$ and all CFC data in GLODAPv2.2021 compared to GLODAPv2.2020. This is due to the removal of data with missing temperatures (Section 3.3.1).

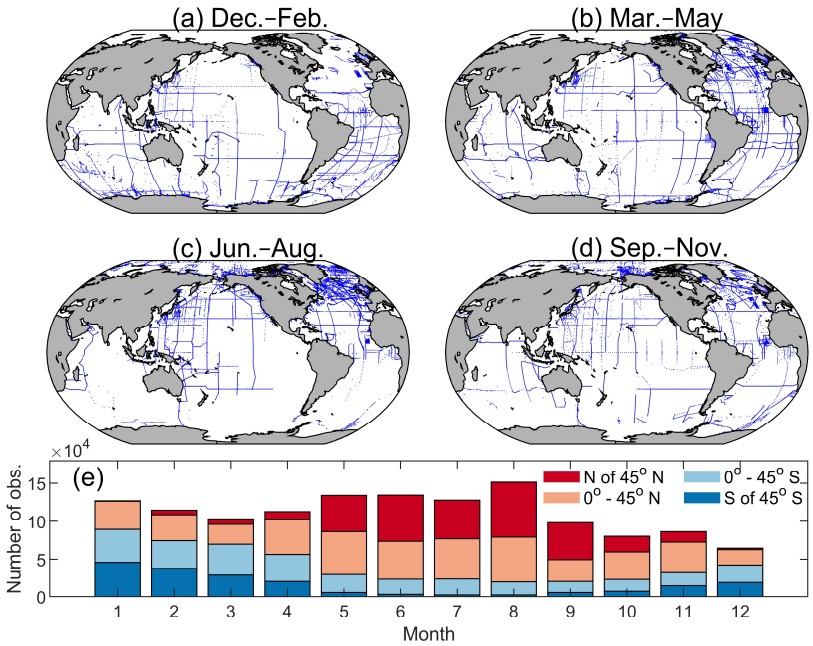

**Figure 9.** Distribution of data in GLODAPv2.2021 in (a) December–February, (b) March–May, (c) June–August, (d) September–November, and (e) number of observations for each month in four latitude bands.

Figure 9 illustrates the seasonal distribution of the data. As for previous versions there is a bias around summertime in the data in both hemispheres; most data are collected during April through November in the Northern Hemisphere while most data are collected during November through April in the Southern Hemisphere. These tendencies are strongest for the poleward regions and reflect the harsh conditions during winter months which make fieldwork difficult. Figure 10 illustrates the distribution of data with depth. The upper 100 m is the best sampled part of the global ocean, both in terms of number (Fig. 10a) and density (Fig. 10b) of observations. The number of observations steadily declines with depth. In part, this is caused by the reduction of ocean volume towards greater depths. Below 1000 m the density of observations stabilizes and even increases between 5000 and 6000 m; the latter is a zone where the volume of each depth surface decreases sharply (Weatherall et al., 2015). In the deep trenches, i.e., areas deeper than ~ 6000 m, both number and density of observations are low.

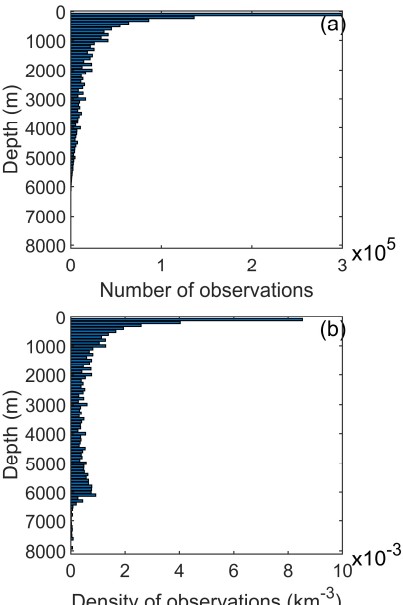

**Figure 10.** Number (a) and density (b) of observations in 100 m depth layers. The latter was calculated by dividing the number of observations in each layer by its global volume calculated from ETOPO2 (National Geophysical Data Center, 2006). For example, in the layer between 0 and 100 m there are on average 0.0075 observations per cubic kilometer. One observation is one water sampling point and has data for several variables.

Except for salinity and oxygen, the core data were collected exclusively through chemical analyses of collected water samples. The data of the 12 core variables were subjected to primary quality control to identify questionable or bad data points (outliers) and secondary quality control to identify systematic measurement biases. The data are provided in two ways: as a set of individual exchange-formatted original cruise data files with assigned WOCE flags, and as globally and regionally merged data product files with adjustments applied to the data according to the outcome of the consistency analyses. Importantly, no adjustments were applied to data in the individual cruise files while primary-QC changes were applied.



The consistency analyses were conducted by comparing the data from the 43 new cruises to the previous data product GLODAPv2.2020. Adjustments were only applied when the offsets were believed to reflect biases relative to the earlier data product release related to measurement calibration and/or data handling practices, and not to natural variability or anthropogenic trends. The Adjustment Table at https://glodapv2-2021.geomar.de/ (last access: 29 June 2021) lists all

applied adjustments and provides a brief justification for each. The consistency analyses rely on deep ocean data (>1500 or 2000 dbar depending on region), but supplementary CANYON-B and CONTENT analyses consider data below 500 dbar. Data consistency for cruises with exclusively shallow sampling was not examined. All new pH data for this version were comprehensively reviewed using crossover analysis, and none required adjustment. Regardless, full reanalysis of all available pH data, particularly in the North Pacific, will be conducted for GLODAPv3.

Secondary QC flags are included for the 12 core variables in the product files. These flags indicate whether (1) or not (0) the data successfully received secondary QC. A secondary QC flag of 0 does not by itself imply that the data are of lower quality than those with a flag of 1. It means these data have not been as thoroughly checked. For $\delta^{13}C$, the QC results by Becker et al. (2016) for the North Atlantic were applied, and a secondary QC flag was therefore added to this variable.

The primary WOCE QC flags in the product files are simplified (e.g., all questionable and bad data were removed). For

salinity, oxygen, and the nutrients, any data flagged 0 are interpolated rather than measured. For $TCO_2$, TAlk, pH, and $fCO_2$ any data flags of 0 indicate that the values were calculated from two other measured seawater $CO_2$ variables. Finally, while questionable (WOCE flag =3) and bad (WOCE flag =4) data have been excluded from the product files, some may have gone unnoticed through our analyses. Users are encouraged to report on any data that appear suspicious.

Based on the initial minimum adjustment limits and the improvement of the consistency resulting from the adjustments

(Table 7), the data subjected to consistency analyses are believed to be consistent to better than 0.005 in salinity, 1 % in oxygen, 2 % in nitrate, 2 % in silicate, 2 % in phosphate, 4 µmol kg$^{-1}$ in $TCO_2$, 4 µmol kg$^{-1}$ in TAlk, and 5 % for the halogenated transient tracers. For pH, the consistency among all data is estimated as 0.01–0.02, depending on region. As mentioned above, the included $fCO_2$ data have not been subjected to quality control, therefore no uncertainty estimate is given for this variable. This should be conducted in future efforts.

**7 Author contributions.**

SKL and TT led the team that produced this update. RMK, AK, BP, SDJ and MKK compiled the original data files. NL conducted the primary and secondary QC analyses. HCB conducted the CANYON-B and CONTENT analyses. CS manages the Adjustment Table e-infrastructure. AK maintains the GLODAPv2 webpages at NCEI/OCADS. PM prepared PYTHON scripts for the merging of the data, and works on converting all code used for the GLODAP effort to

PYTHON. All authors contributed to the interpretation of the secondary QC results and decisions on whether to apply actual adjustments. Many conducted ancillary QC analyses. SKL and AO wrote the manuscript with input from all authors.

**8 Competing interests**

The authors declare that they have no competing interests.



**9 Acknowledgements**

GLODAPv2.2021 would not have been possible without the effort of the many scientists who secured funding, dedicated time to collect, and shared the data that are included. Chief scientists at the various cruises and principal investigators for specific variables are listed in the online cruise summary table. The author team also want to thank the large GLODAP user community for useful input and notification about potential issues in the data products. Such input is invaluable and

helps ensure that GLODAP maintains its high quality and consistency over time. NL was funded by EU Horizon 2020 through the EuroSea action (grant agreement 862626). SKL acknowledges internal strategic funding from NORCE Climate. LCC was supported by Prociencia/UERJ grant 2019-2021. MA was supported by IEO RADIALES and RADPROF projects. PJB was part-funded by the UK Climate Linked Atlantic Sector Science (CLASS) NERC National Capability Long-term Single Centre Science Programme (Grant NE/R015953/1). AV & FFP were supported by

BOCATS2 Project (PID2019-104279GB-C21/AEI/10.13039/501100011033) funded by Spanish Government. RW and BRC acknowledge the NOAA Global Observations and Monitoring Division (fund reference 100007298) and the Office of Oceanic and Atnospheric Research of NOAA. HCB gratefully acknowledges financial support by the BONUS INTEGRAL project (Grant No. 03F0773A). BT was supported through the Australian Antarctic Program Partnership and the Integrated Marine Observing System. MH acknowledges EU Horizon 2020 action SO-CHIC (grant N°821001). We

acknowledge funding from the Initiative and Networking Fund of the Helmholtz Association through the project "Digital Earth" [ZT-0025]. This is CICOES and PMEL contribution numbers 2021-1153 and 5253, respectively. This activity is supported by the International Ocean Carbon Coordination Project (IOCCP).

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





**Appendix A.** Supplementary tables


**Table A1.** Cruises included in GLODAPv2.2021 that did not appear in GLODAPv2.2020. Complete information on each cruise, such as variables included, and chief scientist and principal investigator names is provided in the cruise summary table at  https://www.ncei.noaa.gov/access/ocean-carbon-data-system/oceans/GLODAPv2_2021/cruise_table_v2021.html

| No | EXPOCODE | Region | Alias | Start | End | Ship |
|---|---|---|---|---|---|---|
| 3001 | 06M220140607 | Atlantic | MSM39 | 20140607 | 20140625 | *Maria S. Merian* |
| 3002 | 06M220160331 | Atlantic | MSM53 | 20160331 | 20160509 | *Maria S. Merian* |
| 3003 | 06MT20160828 | Atlantic | M130, SFB754 | 20160828 | 20161003 | *Meteor* |
| 3004 | 06MT20170302 | Pacific | M135, SFB754 | 20170302 | 20170407 | *Meteor* |
| 3005 | 06MT20180213 | Atlantic | M145 | 20180213 | 20180314 | *Meteor* |
| 3006 | 09AR20141205 | Pacific | AU1402 | 20141205 | 20150125 | *Aurora Australis* |
| 3007 | 18DD20100202 | Pacific | LineP-2010-01 | 20100202 | 20100216 | *John P. Tully* |
| 3008 | 18DD20100605 | Pacific | LineP-2010-13 | 20100605 | 20100621 | *John P. Tully* |
| 3009 | 18DD20140210 | Pacific | LineP-2014-01 | 20140210 | 20140224 | *John P. Tully* |
| 3010 | 18DD20150818 | Pacific | LineP-2015-010 | 20150818 | 20150903 | *John P. Tully* |
| 3011 | 18DD20160208 | Pacific | LineP-2016-001 | 20160208 | 20160222 | *John P. Tully* |
| 3012 | 18DD20160816 | Pacific | LineP-2016-008 | 20160816 | 20160831 | *John P. Tully* |
| 3013 | 18DD20160605 | Pacific | LineP-2016-006 | 20160605 | 20160625 | *John P. Tully* |
| 3014 | 18DD20170205 | Pacific | LineP-2017-001 | 20170205 | 20170221 | *John P. Tully* |
| 3015 | 18DD20170604 | Pacific | LineP-2017-006 | 20170604 | 20170620 | *John P. Tully* |
| 3016 | 18DD20190205 | Pacific | LineP-2019-001 | 20190205 | 20190223 | *John P. Tully* |
| 3017 | 18DD20190602 | Pacific | LineP-2019-006 | 20190602 | 20190618 | *John P. Tully* |
| 3018 | 18LU20180218 | Pacific | LineP-2018-001 | 20180218 | 20180308 | *Sir Wilfrid Laurier* |
| 3019 | 18SN20040725 | Arctic | JOIS-2004-16 | 20040725 | 20040802 | *Louis S. St-Laurent* |
| 3020 | 18SN20100915 | Arctic | JOIS-2010-07 | 20100915 | 20101015 | *Louis S. St-Laurent* |
| 3021 | 18SN20110721 | Arctic | JOIS-2011-20 | 20110721 | 20110818 | *Louis S. St-Laurent* |
| 3022 | 18SN20120802 | Arctic | JOIS-2012-11 | 20120802 | 20120830 | *Louis S. St-Laurent* |
| 3023 | 18SN20130724 | Arctic | JOIS2013-04 | 20130724 | 20130902 | *Louis S. St-Laurent* |
| 3024 | 18SN20140921 | Arctic | JOIS-2014-11 | 20140921 | 20141017 | *Louis S. St-Laurent* |
| 3025 | 18SN20160922 | Arctic | JOIS-2016-16 | 20160922 | 20161018 | *Louis S. St-Laurent* |
| 3026 | 18VT20141027 | Pacific | Salish Sea 2014-50 | 20141027 | 20141030 | *Vector* |
| 3027 | 18VT20150401 | Pacific | Salish Sea 2015-17 | 20150401 | 20150405 | *Vector* |
| 3028 | 29AH20090725 | Atlantic | CAIBOX | 20090725 | 20090813 | *Sarmiento de Gamboa* |
| 3029 | 320620170703 | Pacific | GO-SHIP P06W, SOCCOM | 20170703 | 20170817 | *Nathaniel B. Palmer* |
| 3030 | 320620170820 | Pacific | GO-SHIP P06E, SOCCOM | 20170820 | 20170930 | *Nathaniel B. Palmer* |
| 3031 | 320620180309 | Pacific | NBP18_02, SOCCOM | 20180309 | 20180514 | *Nathaniel B. Palmer* |
| 3032 | 325020100509 | Pacific | TN249-10, BEST Spring 2010 | 20100509 | 20100614 | *Thomas G. Thompson* |
| 3033 | 325020190403 | Indian | TN366, GO-SHIP I06S, SOCCOM | 20190403 | 20190514 | *Thomas G. Thompson* |
| 3034 | 33RO20180423 | Indian | GO-SHIP I07N | 20180423 | 20180606 | *Ronald H. Brown* |
| 3035 | 33RR20160321 | Indian | GO-SHIP I09N | 20160321 | 20160428 | *Roger Revelle* |
| 3036 | 35A320031214 | Atlantic | BIOZAIRE III | 20031214 | 20040107 | *L'Atalante* |
| 3037 | 35A320120628 | Pacific | Pandora | 20120628 | 20210806 | *L'Atalante* |
| 3038 | 35A320150218 | Pacific | OUTPACE | 20150218 | 20150304 | *L'Atalante* |
| 3039 | 35MF19820626 | Indian | MEROU-1982-A | 19820626 | 19820703 | *Marion Dufresne* |
| 3040 | 35MF19821003 | Indian | MEROU-1982-B | 19821003 | 19821007 | *Marion Dufresne* |
| 3041 | 49NZ20191229 | Indian | MR19-04, GO-SHIP I07S, | 20191229 | 20200210 | *Mirai* |





| | | | SOCCOM | | | |
|---|---|---|---|---|---|---|
| 3042 | 58JH20190515 | Arctic | JH2019205 | 20190515 | 20190604 | *Johan Hjort* |
| 3043 | 74JC20181103 | Atlantic | GO-SHIP SR01b | 20181103 | 20181123 | *James Clark Ross* |
