# Peer review of "An updated version of the global interior ocean biogeochemical data product, GLODAPv2.2021"

_Earth System Science Data, 2021_

## Author Comment (AC1)

Bergen – November 4, 2021

Dear Editors,

Thank you for considering out manuscript for publication in Earth System Science Data. We greatly appreciate the time and effort you and the reviewers put into the thorough reading and review of this "living data process". Below we include replies to the comments made by the two reviewers.

Sincerely,

Siv K. Lauvset – on behalf of the author team
* * *
Reviewer #1 (Matthew Humphreys)

This manuscript presents the latest update to GLODAP (GLODAPv2.2021) with 43 new cruises added since the previous version along with a few other minor QC-related updates. The changes and additions have been clearly explained and justified where necessary. The explanation of the overall QC procedure (inherited from the previous version of the manuscript) is comprehensive. The QC procedure itself is robust and now well established. **Thank you for these kind words, and for your thorough reading of the updated manuscript.**

I have only a few minor techincal/copy-editing notes and suggestions as follows. Having also been a reviewer for the previous iteration of this manuscript I focused on the track-changes supplement provided by the authors. The line numbers below refer to that version.

185 'two-years' should be 'two years' (no hyphen needed here) **done**

188 'update, is presented here which' should be 'update is presented here, which' **done**

192 remove comma after '989 cruises' **done**

217 'essentially the same' implies that there are minor differences - is this intended? If not, just delete 'essentially'. **Done.**

219-221 why is there no Southern Ocean identifier? Are these cruises included in the Atlantic/Indian/Pacific sectors based on longitude? Please clarify and also mention briefly how the basins are defined (what are the lat/lon/etc. cutoffs). **When beginning work on GLODAPv2 we agreed to have a global product file along with four regional files, and importantly also agreed that there were to be no overlap between these regional files. This decision led to the removal of the Southern Ocean category (which was there in the preceeding CARINA product). Cruises are included in either the Atlantic, Indian, or Pacific based on where most of the data/stations on the cruise are. This is done on a case by case basis and there are no fixed longitudes or cutoffs. This has been clarified in the text. The AMS is generally defined as north of 60°N, but includes the entire Nordic Seas and therefore extends south to the Greenland-Scotland Ridge in the Atlantic Ocean. We agree that this solution for the Southern Ocean is not perfect, and will in future versions allow a cruise to have more than one basin identifier.**

220 should be Arctic 'and' Mediterranean Seas **No, we do mean the Arctic Mediterranean Seas, as in the ocean region north of the Greenland-Scotland Ridge and the Bering Strait. There is no separate basin identifier for the Mediterranean Sea, the few cruises there are identified as Atlantic. This is now noted in the text. This will change in future versions when we add more data from the Mediterranean Sea.**

366 'as the difference, is in accordance' should be 'as the difference, in accordance' **done**

371-372 'The TCO2 data measured on this cruise are high by 2.15 ± 1.04 umol kg-1 when compared to the data measured on nearby cruises' suggest rephrase to 'The TCO2 data measured on this cruise are 2.15 ± 1.04 umol kg-1 greater/higher than the data measured on nearby cruises' **done**

391 'avoid the communicating' should be 'avoid communicating' **done**

425 'pH dependent' should be 'pH-dependent' **done**

425 'in calculated pH with' might be clearer as 'in calculated pH compared with' **done**

428 'pH dependent' should be 'pH-dependent' **done**

518, 525, 529, 530, 532, 533, 534 & 535 missing full stop at end of sentence **fixed**

588 'cruises that ended at 2106' should be 'cruises, which ended at 2106' **done**

615-616 awkward sentence; suggest rephrase 'Neutral density for all 989 cruises was calculated using/following J & M (1997).' **done**

616 missing full stop at end of sentence **done**

740 'Silicate in the Atlantic Ocean, for example, shows' consider rephrase to 'For example, silicate in the Atlantic Ocean shows' **done**

771 https://explore.webodv.awi.de/ link did not work for me (503 Service Unavailable), but it might be a temporary issue **It works when I test on October 22, 2021.**

776 expocode and doi are mostly lowercase but sometimes uppercase as here; suggest to be consistent throughout **The only two expocodes where we use lowercase is "IcelandSea" and "IrmingerSea". These are ship-based timeseries and have these expocodes for historical reasons. It may very well make sense to change these expocodes, but that is something we would prefer to discuss in more detail and rather implement in a future update. The one DOI with uppercase has been changed.**

880 & 881 'Python', not 'PYTHON' **done**

Reviewer #2 (anonymous)

As is usual for GLODAP, the authors have been careful and thorough in their analysis and documentation of this critically important dataset. I have reviewed the manuscript as well as the relatively minor comments from the other reviewer and have no further comments.

**We thank the reviewer for these kind and complementing words.**